

# New Paleogene records of cartilaginous fishes (Chondrichthyes) from central Chile, including the oldest lamnid diversity from the southeastern Pacific

Rodrigo A. Otero

Red Paleontológica U-Chile, Laboratorio de Ontogenia y Filogenia, Departamento de Biología, Facultad de Ciencias, Universidad de Chile, Santiago, RM, Chile
Millennium Nucleus Early Evolutionary Transitions of Mammals, ANID-Milenio, Santiago, RM, Chile
Museo de Historia Natural y Cultural del Desierto de Atacama, Calama, Región de Antofagasta, Chile

Corresponding author
Rodrigo A. Otero,
otero2112@gmail.com

## ABSTRACT

**Background**. The Paleogene chondrichthyan diversity of the southern hemisphere is mostly known in Antarctica, southernmost South America and New Zealand, but records from the southeastern Pacific remain sparse to date. Among these, lamnid sharks (the lineage of the great white shark) are one of the scarcer groups in the southern hemisphere prior to the Eocene; moreover, their occurrences prior to the Neogene remained unreported in the southeastern Pacific. This contribution presents new Paleogene chondrichthyans recovered from two different horizons at Loanco, central Chile, including the first local records of lamnids, with the description of a new species of the genus *Lethenia*.

**Methods**. Sections of two geologic units were studied. These provided new teeth of chondrichthyans, as well as a single associated tooth set plus fragments of jaw cartilage and vertebrae. The material was taxonomically identified and compared with other local occurrences and other coeval assemblages from the southern hemisphere.

**Results**. This research recognizes a lower Paleocene-lower Eocene assemblage including material referable to *Palaeohypotodus* sp., *Megasqualus* sp. and Hexanchidae indet., the latter being the oldest known record in the southwestern Pacific. In addition, material from upper Eocene-lower Oligocene levels include dental pieces referable to the lamnids *Macrorhizodus praecursor* Leriche, and to a new species, *Lethenia carranzaensis* sp. nov., the latter represented by an exceptional specimen preserving a dental set, mandibular cartilage and vertebrae. Although discrete, this fauna shows the presence of Paleocene- lower Eocene elements commonly present in Antarctica and the Austral Basin, suggesting a typical Weddellian distribution during that timespan. On the contrary, the presence of the genus *Lethenia* reinforces the evidence of vertebrate interchange with the north Atlantic between the upper Eocene—lower Oligocene, previously suspected by the shared presence of blochiid billfishes. Evidence of such marine interchange is also present during the late Paleocene, based on the previous reports of *Palaeogaleus*, *Physogaleus*, and *Premontreia*, genera with similar known geographic occurrences. The new records of *Macrorhizodus* and *Lethenia* also represent the oldest known record of Lamnidae in the southwestern Pacific, proving its early presence previous to its widespread and abundant occurrence during the Neogene.

## INTRODUCTION

Paleogene marine vertebrates from the southern Pacific are well-known in Antarctica, New Zealand and the Austral (=Magallanes) Basin, representing a valuable source of paleoenvironmental, paleogeographic and paleobiologic information (*Reguero, Marenssi & Santillana, 2012*; *Reguero et al., 2013*; *Reguero et al., 2022*). The faunal affinities present in different austral localities between the Upper Cretaceous until the Eocene, worked as major support for the Weddellian Biogeographic Province (*Zinsmeister, 1979*; WBP hereafter), a concept initially based on the common distribution of marine invertebrates, later shown to occur also among coeval marine vertebrates of the southern hemisphere (*Reguero, Marenssi & Santillana, 2012*; *Reguero et al., 2022* and references therein). There is consensus about the progressive endemism of the WBP fauna between the Campanian-upper Eocene interval, and along the southern hemisphere (*Zinsmeister, 1979*; *Macellari, 1987*; *Zinsmeister & Macellari, 1988*; *Olivero & Medina, 2000*; *Reguero, Marenssi & Santillana, 2012*; *Reguero et al., 2013*; *Reguero et al., 2022*). In this sense, the local known vertebrate assemblages have provided valuable information in antarctic and subantarctic latitudes, but the coeval record in lower latitudes is still sparse. In particular, the existence of Upper Cretaceous-Paleogene marine units in latitudes 33°–37°S along the coast of central Chile, represents a valuable source of fossil fauna which could allow comparisons to high-latitude coeval assemblages. In this scenario, the Paleogene marine vertebrate records from central Chile become relevant, although, these have been historically disattended and they were mostly represented by material lacking stratigraphic control. Fortunately, in recent years new research allowed fresh insights (*e.g.*, *Muñoz Ramírez et al., 2007*; *Rodriguez, Ward & Quezada, 2023*), but detailed comparisons to known assemblages from other parts of the WBP remain unpracticed.

This contribution presents new discoveries from central Chile, including both upper Paleocene-lower Eocene, and upper Eocene-lower Oligocene chondrichthyan material. The new elements adds to previous records, allowing a first paleogeographic approach. The complemented local chondrichthyan fossil record allows assessing the continuity of the WBP throughout the Paleogene in mid-latitudes of the southeastern Pacific.

## LOCALITY AND GEOLOGICAL SETTING

*Locality*—Loanco is a small cove in the Región del Maule (Maule Region, administrative division of Chile) placed ca. 350 km south from Santiago (Fig. 1A).

*Geologic setting*—In its coast line crops out different sedimentary units deposited during a transgressive-regressive cycle that spanned between the late Campanian to the Eocene/Oligocene boundary. The basement rocks are conformed by Paleozoic metamorphic rocks. From base to roof, the older sedimentary rocks belong to the upper Maastrichtian Quiriquina Formation (*Biró-Bagóczky, 1982*), followed by a small section referable to the Lebu Group (*Cecioni, 1968*), and by a larger section of the Millongue

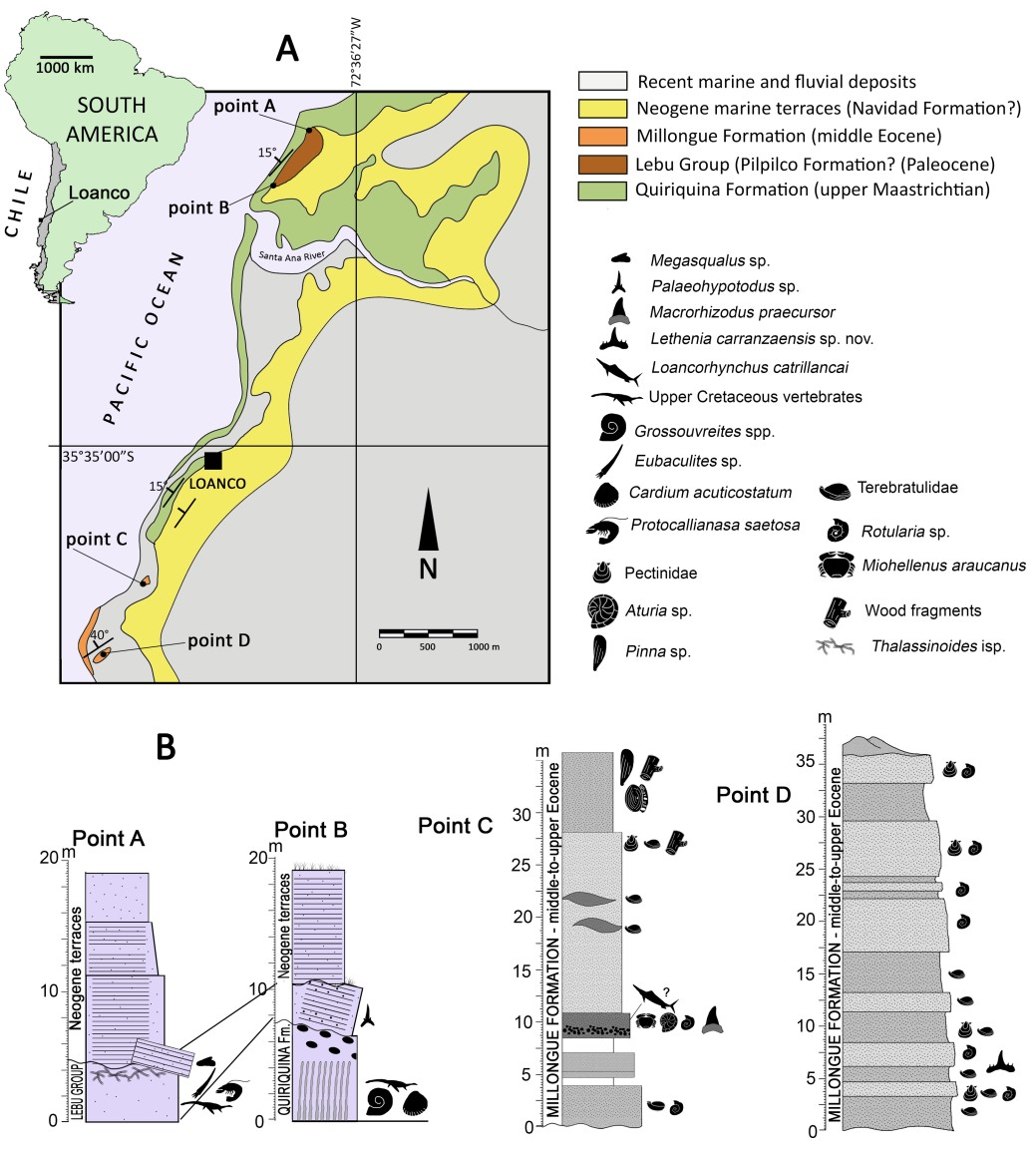

**Figure 1  Area of study.** (A) map indicating the locality of Loanco, in central Chile. Geologic scheme of the outcrops exposed along the coast and cliffs in Loanco. Modified from *Otero (2024)*. (B) Studied stratigraphic sections. Each point is marked on the geologic scheme in (A). Point A and B columns, modified from *Otero (2024)*. C, modified from *Otero (2019)*.

Formation (*Tavera, 1942*). Covering most of these outcrops, there are Neogene marine terraces likely correlated to the Navidad Formation (*Darwin, 1846*). The detailed description of the fossil-bearing units studied is provided as following:

*Lebu Group* (*Cecioni, 1968*; *Le Roux, Nielsen & Henríquez, 2008*)—A small section of soft, yellowish sandstones of variable thickness below five m (Fig. 1B), it overlies through an angular unconformity to a large anticline (ca. 900 m) of the Quiriquina Formation. These levels yielded material referable to the chondrichthyan genus *Palaeohypotodus* (this study), thus, suggesting a Paleocene age. This unit is segregated from the Quiriquina Formation

based on the unconformable contact and different lithology. It is here correlated to the basal levels of the Lebu Group, particularly with the marine Pilpilco Formation (*Muñoz Cristi, 1968*), which represents a first moment of the transgressive cycle. The Pilpilco Formation was regarded as ca. 150 m of fine-grained, greenish sandstones with clay intercalations (*Muñoz Cristi, 1968*). The studied section is consistent with this lithology but much narrower in thickness, suggesting a likely relict of the Pilpilco Formation or else, a different Paleocene marine transgression.

*Millongue Formation* (*Tavera, 1942*; *Muñoz Cristi, 1946*; *Muñoz Cristi, 1973*; *García, 1968*; *Pineda, 1983*)—2.5 km south from Loanco crops out another fossiliferous sedimentary unit characterized by brownish, clayly sandstones of variable grain size (Figs. 1A, 1B). The contact between these levels and those of the Quiriquina Formation is covered by recent sands. Fossil content includes bivalves (*Pinna* sp., Pectinidae), serpulids (aff. *Rotularia* isp.), terebratulid brachiopods, nautiloids (*Aturia* sp.), decapods (*Minohellenus araucanus*), and endemic blochiid fishes (*Otero, 2019*). The lithology and age of the studied outcrops allow correlating them to the middle-to-upper Eocene Millongue Formation (*Tavera, 1942*; *Muñoz Cristi, 1946*).

*Stratigraphic Sections*—*Otero (2024)* provided several Upper Cretaceous stratigraphic sections exposed in different points along the coast close to Loanco, numbered as point 1 to point 5. Two of the stratigraphic sections studied here are equivalent to the points 1 (point A of this study) and the upper levels at point 2 (point B of this study) of *Otero (2024)*. Two additional sampling point are here added, consistently labeled as point C and point D. All these are described as following:

*Point A (35° 33′ 14.1″ S; 72° 36′ 45.8″ W)*— Former point 1 of *Otero (2024)*. The base of this section is covered by recent sands. From base to roof: 4.4 m of a massive clayly sandstone, regularly soft, grey to reddish, with fine to coarse grain and small fossiliferous concretions. The level dips ca. 30° west, being partially overlain by landslides. Fossils include typical vertebrates and invertebrates representative of the Quiriquina Formation (see *Otero, 2024*). In the northern visible flank of the anticline, teeth referable to *Megasqualus* sp. and to Hexanchidae indet. (this study) were recovered. The roof of this level shows frequent traces (*Thalassinoides* isp.) and decapod remains (*Protocallianasa saetosa*). It overlays to levels of the Quiriquina Formation (which conforms the anticline); 7 m of sub-horizontal, grey, compact, hard sandstone with concretions, overlying in angular unconformity. It shows probable bioturbation near its roof; 7 m of grey-to-yellowish, fine sandstone, laminated in its base and having abundant reddish, hard, ferruginous nodules; 5 m of fine compact sandstone, with abundant traces (*Skolithos* isp.).

*Point B (35° 33′ 37.8″ S; 72° 37′ 04.8″ W)*— Former point 2 of *Otero (2024)*. The base of this section belongs to the Quiriquina Formation and is located close to the anticline axis. From base to roof: 6 m of green to grey, hard sandstone with abundant fossiliferous concretions of variable size; four m of poorly consolidated, clayly sandstones, lying in angular unconformity over the anticline. Near its base, there are reworked fossils (actinopterygian scales and fragments *Pacitrigonia hanetiana*). A single *in situ* tooth was recovered from this level (referred to *Palaeohypotodus* sp. in this study); 5 m grey, unconsolidated, fine sandstones without fossils.

*Point C (35° 33′ 46.8″ S; 72° 37′ 53.8″ W)*— The base of this section is covered by recent sands. Three m of greenish sandstones with serpulid banks; 0.5 m with no outcrop; two m of reddish sandstones; two m with no outcrop; 1.8 m of brown mid-grained sandstone with micaceous fragments, and having a fine conglomerate near its base. Fossils in this level include teeth of *Macrorhizodus praecursor* (this study), a phragmocone of the nautiloid *Aturia* sp. (SGO.PI.6776), and an articulated individual of *Minohellenus* ('Imaizula') *araucanus*; 14.3 m of mid-grained, yellow-to-green sandstone with lenses including banks of Terebratulidae indet.; pectinids and carbonized wood fragments occur near the roof of the level; six m of mid-grained, reddish to brown sandstones, with several *in situ* individuals of *Pinna* sp. in life position. This level also includes an isolated lamniform vertebra (*in situ*), serpulid traces, carbonized wood fragments, and *Teredolites* isp.

*Point D (35° 36′ 12.39° S; 72° 38′ 13.26″ W)*— Following *Valdés (2019)*, this section comprises ca. 30 m of quartz sandstones, intercalated by brownish and green sandstones with few glauconitic levels. The section is very fossiliferous, diminishing in abundance from base to roof. Grain size increases from base to roof. The upper part of the exposed section becomes orange near the roof. The brownish levels are comparatively prominent with respect to the green sandstones, showing better resistance to the tidal erosion. Most of the brownish levels lack bivalves, but contains frequent traces of calcareous worms (*Rotularia* sp.) and occasional terebratulid brachiopods. On the contrary, green levels include bivalves, terebratulids, and marine vertebrates (associated chondrichthyan teeth, SGO.PV.6635).

*Age of the studied outcrops*—The presence of *Palaeohypotodus* (with well-preserved crown and cusplets) on the younger unit (tentatively correlated to the Pilpilco Formation at points A and B), suggests a Paleocene age based on the known biochron of this genus (*Cappetta, 2012*), although austral occurrences of *Palaeohypotodus* are also known in the lower Eocene of southernmost Chile (*Otero & Soto-Acuña, 2015*) and the middle Eocene of Antarctica (*Long, 1992*; *Kriwet, 2005*; *Reguero et al., 2013*; *Kriwet et al., 2016*; *Charnelli et al., 2024*), thus, supporting a local Paleocene-Eocene age.

The younger unit (Millongue Formation, points C and D) crops out in the southern part of the studied locality. Its eventual contact with the Quiriquina Formation is obscured by recent coastal deposits. The presence of *Aturia* sp. and *Minohellenus araucanus* constrain the unit to an Eocene age, suggesting an eventual extension into the Oligocene (*Nielsen, Bandel & Kröger, 2006*; *Schweitzer & Feldmann, 2002*; *Schweitzer et al., 2010*). In addition, the local presence of blochiid fishes (*Otero, 2019*) suggests a middle-to-late Eocene age based on the known biochron of blochiids (*Fierstine & Monsch, 2002*). The relative age of these outcrops, their distinctive lithology and marine environment, support a correlation with the Millongue Formation (*Tavera, 1942*), which is present in southern and northern localities of the Arauco Basin (*Tavera, 1942*; *Muñoz Cristi, 1968*; *Tavera, 1980*). The age of the Millongue Formation was independently constrained to the middle-to-late Eocene based on fossil invertebrates in other localities of central Chile (*Tavera, 1942*; *Tavera, 1980*). The faunal assemblage locally recognized in Loanco (points C and D) indicates a late Eocene or even early Oligocene age for these outcrops.

## MATERIALS & METHODS

*Institutional abbreviations*—SGO.PV., Área Paleontología, Museo Nacional de Historia Natural, Santiago, Chile.

*Material*—The studied material was collected in several fieldworks. The specimen SGO.PV.6635 was collected by the author on 2008, and subsequently prepared by hand tools. Natural crackings of the original sandstone block affected several teeth. These were separated from the matrix and later re-joined with cyanocrilate, although the contact surfaces were damaged in few cases. Few teeth are preserved in hardened nodules. Then, their preparation was risky considering the presence of very delicate lateral cusplets.

The remaining teeth were recovered as isolated elements during successive field works between 2010 and 2015, being later separated from its hosting matrix by hand tools (fine chisels, dental instruments).

*Nomenclatural Acts*—The electronic version of this article in Portable Document Format (PDF) will represent a published work according to the International Commission on Zoological Nomenclature (ICZN), and hence the new names contained in the electronic version are effectively published under that Code from the electronic edition alone. This published work and the nomenclatural acts it contains have been registered in ZooBank, the online registration system for the ICZN. The ZooBank LSIDs (Life Science Identifiers) can be resolved and the associated information viewed through any standard web browser by appending the LSID to the prefix http://zoobank.org/. The LSID for this publication is: urn:lsid:zoobank.org:pub:FF2D99EE-7705-4F96-A7F4-53D18A520EF9.

The online version of this work is archived and available from the following digital repositories: PeerJ, PubMed Central SCIE and CLOCKSS.

### Systematic paleontology

Chondrichthyes *Huxley, 1880*
Elasmobranchii *Bonaparte, 1838*
Neoselachii *Compagno, 1977*
Squaliformes *Compagno, 1973*
Squalidae *De Blainville, 1816*
Genus *Megasqualus Herman, 1982a*; *Herman, 1982b*

*Type Species*—'*Notidanus orpiensis*' *Winkler, 1874*, upper Paleocene of Belgium.
*Megasqualus* sp.
(Figs. 2A–2B)

*Centrophoroides* sp.: *Suárez & Otero, 2008*
*Centrophoroides* sp.: *Otero, 2015*.
*Referred Material*—SGO.PV.6625, a single lateral tooth.
*Locality and Horizon*—Loanco, Point A, basal marine unit of the Lebu Group (Pilpilco Formation), Paleocene-lower Eocene.
*Description*—Squalid large tooth (6 mm wide) with a crown strongly recurved backwards, having a nearly straight cutting edge with few irregular slight serrations over its mesial

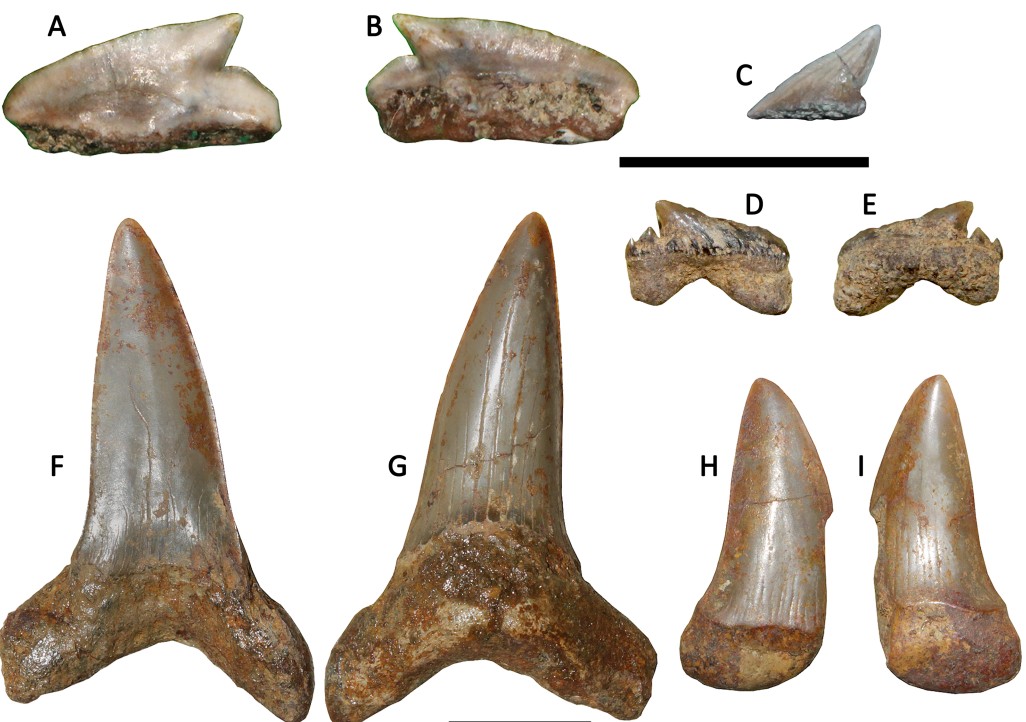

**Figure 2 Chondrichthyans from Loanco, central Chile.** (A) *Megasqualus* sp. (SGO.PV.6625), isolated lateral tooth in labial view. (B) Same in lingual view. (C) Hexanchidae indet. (SGO.PV.6626) isolated main crown in lingual view. (D) *Palaeohypotodus* sp. (SGO.PV.6780), single posterior tooth. Pilpilco Formation, upper Paleocene-lower Eocene. (F) *Macrorhizodus praecursor* (*Leriche, 1905*) (SGO.PV.6633a, referred) upper inferior tooth in labial view. (G) Same in lingual view. (H) *Macrorhizodus praecursor* (*Leriche, 1905*) (SGO.PV.6633b, referred), incomplete tooth in labial view. (I) Same in lingual view. Millongue Formation, upper Eocene-lower Oligocene. Scale bar equals 10 mm in both cases.

margin. The crown has slightly protruding uvula. The labial apron is broken, but its contour is preserved. This shows a basal projection that does not reach the root base. The root base is concave and it has a rhomboidal outline. There are no mesial or distal hollows, indicating that the dental overlap is weak in this taxon.

*Comparisons*—SGO.PV.6625 was initially referred to as *Centrophoroides* sp. by *Suárez & Otero (2008)*. After additional preparation, the basal projection of the labial apron was revealed. The latter is basally projected but not extended beyond the root base. Among Squalidae (*i.e.,* genera *Centrophoroides, Centrosqualus, Megasqualus, Protosqualus,* and *Squalus*), the tooth size is informative. According to *Cappetta (2012)*, the genus *Centrosqualus* is characterized by teeth up to 2 mm, *Squalus* reaches up to 5 mm, while *Centrophoroides* and *Protosqualus* reach less than 7 mm. Morphologically, SGO.PV.6625 differs from *Centrosqualus* and from *Squalus.* Both genera have a long, basally projected root (*Cappetta, 2012*: figs. 99 and 102), contrary to the basally short root and brief apron of the Chilean specimen. On the other hand, SGO.PV.6625 morphologically resembles to *Centrophoroides, Protosqualus* and *Megasqualus,* but differs from *Centrophoroides* by lacking a concave basal face of the root and by their comparatively larger size (*Cappetta, 2012*).

Otherwise, SGO.PV.6625 shows a straight root base, differing from that of *Protosqualus* which is characterized by a basal face forming an obtuse angle with the plane of the crown (*Cappetta, 2012*: p. 114). SGO.PV.6625 is thus referred to as *Megasqualus* based on its large size close to 1 cm, its flat basal face of the root, and by its apicobasally root height shorter than the crown.

SGO.PV.6625 differs from the typical local teeth of the Arauco Basin referred to *Centrophoroides*, which have a large apron that basally surpasses the entire root (*Otero, 2024*). The crown cutting edge of SGO.PV.6625 is almost straight, also differing from the local records referable to *Centrophoroides*, which have comparatively higher crowns with irregularly serrated cutting edges, and, in some cases, a differentiated mesial cutting edge with fine serrations and a distal cutting edge with a higher angle and more marked irregular serrations, resembling a separation between the lateral cusp and the crown (*Otero, 2024*).

The Upper Cretaceous austral record of Squaliformes includes similar ecomorphotypes. Particularly, *Protosqualus argentinensis Bogan, Agnolin & Novas (2016)* was described from upper Maastrichtian beds of Argentinean Patagonia. Teeth of this taxon have unusually irregular serrations, a high triangular crown posteriorly recurved, and a triangular, lingually projected uvula. These traits are not present in SGO.PV.6625. Moreover, the low crown highly recurved backwards, the shallowly projected apron and the slight serrations are characters described for the genus *Megasqualus* (see *Cappetta, 1987*: p. 55). SGO.PV.6625 is slightly larger than the *Centrophoroides* teeth frequently found in the Arauco Basin, which rarely exceeds 5 mm in length (*Otero, 2024*).

*Remarks*—The genus *Megasqualus* has been reported in the upper Paleocene-lower Eocene of Belgium and England (*Gurr, 1962*; *Herman, 1982a*; *Herman, 1982b*; *Cappetta, 1987*; *Cappetta, 2012*), and the lower-middle Paleocene of New Zealand (*Mannering & Hiller, 2008*).

Hexanchiformes *De Buen, 1926*
Hexanchidae *Gray, 1851*

Hexanchidae indet.
(Fig. 2C)

cf. *Echinorhinus* sp.: *Suárez & Otero, 2008*; *Otero, 2015*.
*Referred Material*—SGO.PV.6626, an isolated crown fragment.
*Locality and Horizon*—Loanco, Point A, basal marine unit of the Lebu Group (Pilpilco Formation), Paleocene-lower Eocene.
*Description*—Triangular cusp strongly recurved backwards, with complete cutting edges. Its labial face is flat and the lingual face convex. The preserved cusp bears basal wrinkles and few longitudinal striations over the enameloid. Over the cutting edges, small radial striations are visible. A small fragment of the root is attached to the cusp base, which allows delimiting the complete cusp height.
*Comparisons*—This cusp was previously considered to be related to the genus *Echinorhinus* (*Suárez & Otero, 2008*). However, the presence of a convex lingual face differs from the very flat crown present in homodont teeth of *Echinorhinus* (*Cappetta,*

*1987*; *Cappetta, 2012*). Even more, the longitudinal striations over the lingual face of the cusp and the radial striations associated to the cutting edges are features seen in hexanchids such as *Notidanodon* (see *Bogan, Agnolin & Novas, 2016*: fig. 2B). The isolated cusp SGO.PV.6626 is not enough for a generic identification, reason why it is kept as an indeterminate hexanchid.

*Remarks*—The eventual presence of hexanchids in pre-Neogene units of central Chile was first regarded by *Philippi (1887)* who referred to *Notidanus* few teeth from the lower Maastrichtian of Algarrobo, central Chile. Auspiciously, the material was figured by Philippi (*1887*: Plate 55, figs. 11A, 11B), revealing that these indeed belong to teeth of squaliforms (likely *Centrophoroides*), thus, discarding its adscription to hexanchids.

Previous to this research and considering the updated information, hexanchids were known in central Chile from the Miocene and onwards. Thus, SGO.PV.6626 is the first and to date the unique evidence of this clade in the Paleocene-lower Eocene of the southeastern Pacific. On the contrary, WBP hexanchids have been frequently reported in the upper Campanian-lower Maastrichtian deposits of James Ross Island, Antarctica (*Richter & Ward, 1990*), the upper Maastrichtian of Seymour Island, Antarctica (*Grande & Chatterjee, 1987*) and the upper Maastrichtian of the Austral Basin (*Bogan, Agnolin & Novas, 2016*). During the Paleogene, hexanchids have been reported in all the main localities within the WBP (*Cione & Reguero, 1994*; *Kriwet, 2005*; *Mannering & Hiller, 2008*; *Reguero, Marenssi & Santillana, 2012*; *Reguero et al., 2022*; *Otero et al., 2012*; *Otero et al., 2013*; *Kriwet et al., 2016*), except the southeastern Pacific.

Lamniformes *Berg, 1958*
Odontaspididae *Müller & Henle, 1838*
Genus *Palaeohypotodus* Glickman, 1964

*Type species*—'*Odontaspis*' (=*Palaeohypotodus*) *rutoti* (*Winkler, 1874*). Late Paleocene of Belgium.

*Palaeohypotodus* sp.
([Figs. 2D](), [2E]())

Triakidae indet.: *Otero, 2015*.
*Referred Material*—SGO.PV.6780, a single posterior tooth.
*Locality and Horizon*—Loanco, point B, basal marine unit of the Lebu Group (Pilpilco Formation), Paleocene-lower Eocene.
*Description*—Posterior tooth with a low triangular crown which is flat on the labial face and convex in the lingual face. Its labial face has strong folds that fade into the tip, while the lingual face has a smooth enameloid. Labially, the crown overhangs to the root by a bulge. The distal side has two triangular and robust cusplets, and the mesial side lacks any cusplet. The root has two separated and thick lobes with a shallow medial nutritious groove.
*Comparisons*—This tooth was first referred by *Otero (2015)* as an indeterminate triakid, without providing its repository neither figuring it. Additional preparation revealed the presence of a triangular crown, two distal cusplets and absence of mesial cusplets. Such features resemble teeth of several Carcharhiniformes (*e.g.*, the genus *Galeorhinus*). However,

the root has two prominent, divergent lobes and a shallow medial nutritious groove, differing from all Carcharhiniformes (*i.e.,* Scyliorhinidae, Proscyllidae, Pseudotriakidae, Triakidae and Carcharhinidae), which possess deep medial grooves, and a medial part of the root thicker than the root lobes, the latter being rarely divergent (*Cappetta, 2012*). Instead, divergent lobes medially separated by a nutritious groove are typical traits among Odontaspididae (*Cappetta & Nolf, 2005*). The lack of mesial cusplets in SGO.PV.6780 and its low, strongly recurved cusp, are consistent with a very posterior tooth position. In addition, the two distal cusplets are both triangular and they are medially recurved.

*Cappetta (2012)* included in Odontaspididae the genera *Araloselachus, Borealotodus, Brachycarcharias, Carcharias, Cenocarcharias, Glueckmanotodus, Hispidaspis, Hypotodus, Jaekelotodus, Johnlongia, Mennerotodus, Odontaspis, Orpodon, Palaeohypotodus, Roulletia, Sylvestrilamia,* and *Turania.* The presence of profuse and strong labial folds between the crown and the root is a feature only present in the odontaspidid genera *Cenocarcharias, Johnlongia, Palaeohypotodus,* and occasionally, in *Orpodon* (*Cappetta, 2012*: figs. 184, 190, 192, 193). The presence of triangular cusplets in SGO.PV.6780 differs from the sharp and medially recurved cusplets present in *Johnlongia* (*Cappetta, 2012*: figs. 190P–190U). In addition, the non-continuous cutting edge between the lateral cusplets and the cusp in SGO.PV.6780 differs from the continuous cutting edge described for *Cenocarcharias* (*Cappetta & Case, 1999*). Moreover, the latter genus is restricted to the Cenomanian, thus precluding a possible occurrence due to the lack of coeval beds in the currently studied area. Comparison of SGO.PV.6780 with *Orpodon* shows noticeable differences in the larger size of the first (*Orpodon* does not exceeds 12 mm in anterior teeth; see *Cappetta, 2012*: p. 205), and particularly, the posterior teeth of *Orpodon* usually lack a secondary distal cusplet, while the occurrence of basal folds in the crown is occasional (*Cappetta, 2012*: p. 205). On the contrary, all the morphologic traits described above for SGO.PV.6780 (*i.e.,* The separated root lobes, the labial bulge separating the crown and the root, and the strong labial enameloid folds) are features present in the odontaspidid genus *Paleohypotodus* (*Cappetta, 1987*; *Cappetta, 2012*). Coincidently, the genus *Palaeohypotodus* counts with several coeval records previously reported in the austral Pacific (see Discusion chapter).

Lamnidae (*Müller & Henle, 1838*)
Genus *Macrorhizodus Glikman, 1964*

*Type Species*—*Isurus praecursor* (*Leriche, 1905*). Middle Eocene, Belgium.

*Macrorhizodus praecursor* (*Leriche, 1905*)
(Figs. 2F–2I)

*Macrorhizodus praecursor*: *Otero, 2015*
*Referred Material*—SGO.PV.6633. Two complete anterolateral teeth.
*Locality and Horizon*—Loanco, Point C. Millongue Formation, upper Eocene-lower Oligocene.
*Description*—The larger tooth has a high triangular crown with complete, non-serrated cutting edges. The labial face is flat, while the lingual face is convex. The root has two slightly divergent lobes with a squared contour. Its general shape suggests a lower anterior

position. The smallest tooth has similar features. Its crown is comparatively short and mesiodistally larger, with the tip recurved backwards, evidencing its lateral position. The mesial side of the tooth is broken. The root, the cutting edges and the enameloid show signs of erosion, indicating transportation prior its burial.

*Comparisons*—Eocene teeth of this lamnid morphotype found in Antarctica have been first ascribed to the genus *Isurus* (*Cione & Reguero, 1994*; *Reguero, Marenssi & Santillana, 2012*; *Kriwet et al., 2016*). However, the genus *Isurus* has both fossil and extant species (*i.e., Isurus oxyrhincus* and *Isurus paucus*) with teeth much different possessing slender and sigmoidal crowns, and roots with divergent lobes (*Compagno, 2001*). In addition, the cutting edge is complete in teeth of *Macrorhizodus* whereas in teeth of *Isurus* it generally does not reach the base (*Cappetta, 2012*; *Carlsen & Cuny, 2014*). Similar lamnid teeth have been found in two middle-to-late Eocene localities of southernmost Chile, being referred to *Macrorhizodus praecursor* based on the differences listed above (*Otero et al., 2012*; *Otero et al., 2013*), but also considering their similarity to Neogene taxa such as *Carcharodon hastalis* and *Carcharodon carcharias* (*Ehret et al., 2012*), characterized by large, triangular upper anterior teeth. Following these criteria, the new specimens from central Chile are also referred to *Macrorhizodus praecursor*.

Genus *Lethenia Baut & Génault, 1999*

*Type Species*—'*Odontaspis Van der Broecki*' (*Winkler, 1880*; p. 77-78), lower Oligocene of Limburg, Belgium.

*Lethenia carranzaensis* sp. nov.
lsid:zoobank.org:act:012B0463-22D3-4A40-916A-F005532BD09B
(Figs. 3A–3Q)

*Isurolamna* sp.: *Suárez & Otero, 2010*
*Isurolamna* sp.: *Otero, 2015*
*Isurolamna* sp.: *Otero, 2019*

*Holotype*—SGO.PV.6635. Associated remains of a single specimen including five anterior or near anterior teeth, four lateral teeth, one posterior tooth, two vertebrae and fragments of palatoquadrate and Meckel's cartilage.

*Locality and Horizon*—Loanco, Point D. Millongue Formation, upper Eocene-lower Oligocene.

*Etymology*—After the toponymy of Carranza lighthouse, north Loanco, Región del Maule, central Chile.

*Diagnosis*—Species within *Lethenia* having two blunt triangular lateral cusplets in anterior, lateral and posterior teeth with complete cutting edges reaching the base of the crown, nearly subrectangular root lobes, especially in anterior and anterolateral teeth, and straight lateral cusplets. On the contrary, *Lethenia vanderbroecki* has incomplete cutting edges, lingually and laterally recurved cusplets, rounded root lobes and curved lateral cusplets.

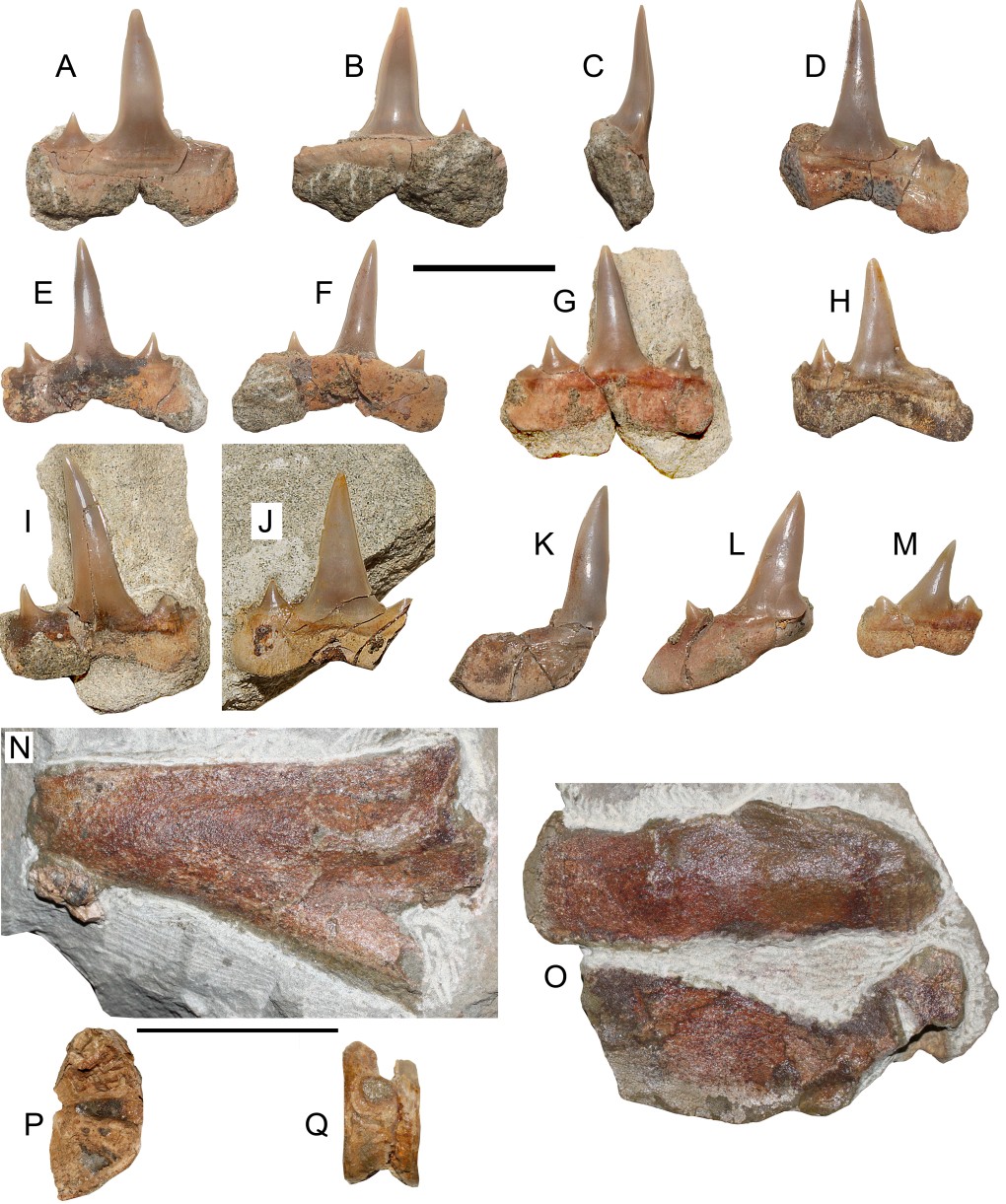

**Figure 3** *Lethenia carranzaensis* **sp. nov. (SGO.PV.6635, holotype).** *Lethenia carranzaensis* sp. nov. (SGO.PV.6635, holotype). Preserved teeth. (A) Anterior tooth in labial view. (B) Same in lingual view. (C) Profile view. (D) Incomplete anterior tooth. (E) Anterolateral tooth in labial view. (F) Same in lingual view. (G) Anterior tooth in labial view. (H) Anterolateral tooth in labial view. (I) Posterolateral tooth in labial view. (J) Anterior tooth in labial view. (K, L, M) Different posterolateral teeth. (N) Symphyseal fragment of the Meckel's cartilage (left ramus). (O) Posterior fragments of palatoquadrate and Meckel's cartilage (both left rami) in relative occlusion. (P) Associated vertebra in articular view. (Q) Same in lateral view. Scale bars equals 10 mm, except N–Q, 50 mm.

*Description*—Anterior teeth with straight cusp in labial/lingual and profile views, with complete cutting edges. The main cusp has a D-shaped section, without ornamentation. There are two lateral cusplets with triangular outline on each side of the main cusp. The

larger cusplet is triangular, relatively low and broad, having a well-formed apex and it has complete cutting edges. The base of the larger cusplet is detached from the main cusp. The smallest cusplets are mesially and distally placed. These are triangular but having rounded tips. The root is mesiodistally larger than the main cusp height. Roots have a slightly squared contour without divergent lobes. Labiolingually, the roots are thin, without any bulk or projection. Three directly associated cartilage fragments were recovered from the same block. These comprise a symphyseal portion of the Meckel's cartilage (left ramus), and two posterior fragments of the Meckel's cartilage and palatoquadrate in occlussion. A single, mostly eroded and fragmentary vertebra was also recovered.

*Comparisons*—SGO.PV.6635 is referred to the Lamnidae based on the distinctive flat root with squared lobes, coincident with those of extant lamnid genera (*i.e., Carcharodon, isurus* and *Lamna*) and also present in several extinct genera (*i.e., Cosmopolitodus, Isurolamna, Karaisurus* and *Lethenia*: see *Cappetta, 2012*: figs. 199–205).

*Cappetta (2012)* included among Lamnidae, the genera *Carchariolamna, Carcharodon, Carcharoides, Cosmopolitodus, Isurolamna, Isurus, Karaisurus, Lamna, Lethenia* and *Macrorhizodus.* The upper and lower anterior teeth of *Carcharodon, Cosmopolitodus* and *Macrorhizodus* are characterized by high cusps (broad triangular in upper anterior teeth, and sharp, comparatively narrower in lower anterior teeth) without lateral cusplets. SGO.PV.6635 anterior teeth show the presence of one or two lateral cusps, a condition shared with the genera *Carcharoides, Isurolamna, Isurus, Karaisurus, Lamna,* and *Lethenia.* However, *Carcharoides, Karaisurus, Isurolamna* and *Lamna* possess all high triangular cusps, contrary to the narrow and slender cusps of SGO.PV.6635, which are only shared with those of the genus *Lethenia.*

The originally monotypic genus *Lethenia Baut & Génault (1999)* included only the species *Lethenia vanderbroecki* (*Winkler, 1880*). This taxon was diagnosed by the presence of low lateral cusplets without any ornamentation and well-separated from the crown, with sub-circular cross-section, plus very compressed, divergent root branches. SGO.PV.6635 shares with *Lethenia vanderbroecki* the unornamented cusplets separated from the crown, as well as the distinctive compressed shape of the root branches. On the contrary, it differs from the latter species by possessing two lateral cusplets on each side, and by having cusplets comparatively blunter, with a triangular outline, contrary to the sharp and thin cusplets of *Lethenia vanderbroecki* (*Winkler, 1880*: p. 77; *Baut & Génault, 1999*: fig. 11; *Nolf, 1988*: pl. 47). While *Lethenia vanderbroecki* was originally described based on twenty complete but isolated teeth, the Chilean specimen represents a partial dental set of a single individual, plus a few other associated skeletal remains. This set allows assessing the morphology of the anterior, lateral and posterior teeth. All the teeth similarly possess two cusplets on each side of the crown (this can be assessed in the incomplete teeth based on the basal section of the missing cusplets) On posterior-most teeth, the crown and the cusplets appear as immediately adjacent elements without an evident separation, differing from the separated crown and cusplets considered as a diagnostic trait of *Lethenia* (*Baut & Génault, 1999*: p. 23). However, the specimens figured by *Nolf* (*1988*: pl. 467, Figs. 2 and 3) show its crown and cusplets with a complete contact between them. Thus, the variable separation between the crown and cusplets could be related to the individual tooth growth, being presumably

contacted in recently replaced teeth, but acquiring a marked separation in old, mature teeth. If this is the case, this trait should be abandoned as part of the diagnosis of the genus.

*Remarks*—Previous to this research, SGO.PV.6635 was considered to be related to the genus *Isurolamna Cappetta (1976)*, mostly based in the non-divergent root branches observed in the first teeth recovered from the matrix during its preparation (*Suárez & Otero, 2010*; *Otero, 2015*; *Otero, 2019*). However, SGO.PV.6635 dental set shows that the crown of all teeth is straight, contrary to the sigmoidal profile regarded in the diagnosis of *Isurolamna* (*Cappetta, 1976*; *Cappetta, 1987*). SGO.PV.6635 also lacks a marked heterodonty (as described for *Isurolamna* by *Cappetta, 1976*), instead of having a low heterodonty, mostly relying on the posterior curvature of the crown of lateral and posterior teeth. Based on the available set, there is also no evidence of a dignathic heterodonty on SGO.PV.6635. Moreover, the crowns of the known species within *Isurolamna* (*i.e., Isurolamna barajunasi Glikman & Zhelezko, 1985*; *Isurolamna inflata Leriche, 1905* and *Isurolamna gracilis Le Hon, 1871*) are high and sigmoidal, comparatively blunter in the case of *I. gracilis* and *I. barajunasi*, and the roots show divergent branches in the three species. On the contrary, all the material historically referred to the genus *Lethenia*, shows straight crowns and non-divergent root branches. While the species *Lethenia vanderbroecki* has been included within *Isurolamna* by some authors (*e.g.*, *Dutheil, 1991*), the traits discussed by *Baut & Génault (1999)* for supporting *Lethenia* as a taxon different from *Isurolamna,* are verified here on the basis of the first available associated dental set including anterior, lateral and posterior elements of a single individual. Considering these facts, this research proposes a second species within *Lethenia*, being its first occurrence in the southern hemisphere. On contrary, the dental features of *Isurolamna* are still based on artificial sets (*Le Hon, 1871*; *Cappetta, 1976*; *Zhelezko & Kozlov, 1999*; *Kovalchuk et al., 2023*) and likely, the genus may represent a wastebasket taxon.

## DISCUSSION

*Previous records of Paleogene chondrichthyans in central Chile*—Earliest reports include the mention of *Odontaspis elegans* from the Eocene of central Chile (*Oliver-Schneider, 1936*), although, the material was never figured, and its repository is currently unknown. *Suárez & Marquardt (2003)* referred to 'Myliobatis sp.' several plates from the Eocene of Algarrobo, but their repository was not provided and the specimens were not figured, making impossible its reassessment. Later, *Muñoz Ramírez et al. (2007)* described a diverse chondrichthyan assemblage from Talcahuano (ca. 135 km south from the localities studied here) considering them as part of the Late Cretaceous Quiriquina Formation (*Biró-Bagóczky, 1982*), and including *Squalus* sp., *Squatina* sp., *Cretorectolobus* sp., *Carcharias* sp., *Palaeohypotodus* sp., *Scapanorhynchus* sp., *Palaeogaleus* sp., *Galeorhinus* sp., *Paraorthacodus* sp., *Dasyatis* sp., *Rhinoptera* sp., and ?Dasyatidae indet. *Suárez & Otero (2010)* described the presence of *Isurolamna* sp. on Loanco, Región del Maule, which is the material here described as a new species of *Lethenia*. After, *Groz & Palma-Heldt (2013)* reviewed the assemblage described by *Muñoz Ramírez et al. (2007)*, recognizing that several known biochrons were indeed restricted to the Paleocene. *Suárez (2015)* reviewed (among others)

the Paleogene record from central Chile, listing all these previous records. *Otero (2015)* mentioned the presence of *Macrorhizodus praecursor* in the middle-to-upper Eocene of Loanco. The specimens were not figured nor repository was indicated then. These are described here for the first time. Later, *Fernández-Jiménez et al. (2016)* reported about additional chondrichthyans referred to *Carcharias 'hopei'*, *Carcharias* sp., *Striatolamia macrota*, and *Myliobatis* sp., reinforcing the presence of Paleocene-Eocene levels at Talcahuano.

More recently, *Rodriguez, Ward & Quezada (2023)* reported additional late Paleocene chondrichthyans from Talcahuano in central Chile, with material referred to *Paraorthacodus clarkii*, *Squalus minor*, *Squalus orpiensis*, *Centrophorus* sp., *Squatina prima*, *Anomotodon novus*, *Striatolamia striata*, *Carcharias* spp., *Sylvestrilamia teretidens*, *Odontaspis winkleri*, *Palaeohypotodus speyeri*, *Palaeohypotodus rutoti*, *Isurolamna inflata*, *Premontreia gilberti*, *Physogaleus secundus*, *Palaeogaleus vincenti*, and *Hypolophodon sylvestris*. It must be observed that this diversity needs a critical review. The identification of three squalid taxa (*i.e., Squalus minor*, *Squalus orpiensis* and *Centrophorus* sp.) was each based on a single tooth. Squalidae are characterized by a poor monognathic and dignathic heterodonty, meaning that teeth from the upper and lower jaw are similar and the teeth of each jaw are also similar in different dental positions (*Cappetta, 2012*: fig. 12C). Moreover, teeth of different squalid genera can reach similar general shape during individual tooth growth. Because of this, the usage of isolated squalid teeth is problematic for granting specific determinations. A similar problem occurs among Odontaspididae and Mitsukurinidae. These groups have marked dignathic dentition and they also have a marked dental variation along each jaw; however, different genera can show remarkably similar teeth depending on their dental position or in the jaw (*Cunningham, 2000*; *Cappetta & Nolf, 2005*). For preventing these issues, a large sampling is recommended and has proved to work on other local fossil odontaspidids and squalids (*e.g.*, the Late Cretaceous record of *Carcharias gracilis* was based on 196 teeth from the same locality and level; see *Otero, 2024*). On the other hand, part of the specimens of *Rodriguez, Ward & Quezada (2023)* are reported to be abraded (*i.e., Centrophorus* sp., *Squatina* sp.), leaving the possibility of reworked material from older strata. Both squalid and squatinid sharks are known in local older underlying units such as the Quiriquina and Cosmito formations (*Suárez et al., 2003*; *Muñoz Ramírez et al., 2007*; *Groz & Palma-Heldt, 2013*).

With these considerations, the present contribution recommends that the specific identifications of squalids, mitsukurinids and odontaspidids based in a single sample as provided by *Rodriguez, Ward & Quezada (2023)*, should kept under open nomenclature awaiting larger samplings that could support their specific adscription, with the exception of *Striatolamia striata* and *Hypolophodon sylvestris* which are supported by a fair number of dental pieces. On the other hand, the records of *Sylvestrilamia teretidens* ($n = 1$; *Rodriguez, Ward & Quezada, 2023*: figs. 7N, 7O) and *Isurolamna inflata* (based on $n = 1$; *Rodriguez, Ward & Quezada, 2023*: figs. 8C, 8D) likely represent posterior teeth of other odontaspidids and are difficult to assure even to genus level based on a single available tooth.

A summary of the Paleogene chondrichthyan records from central Chile with remarks on their status (after the comments above), is provided in Table 1.

**Table 1** **Summary of the Paleogene chondrichthyan record from central Chile.** Available publications, identified taxa and provenance, ordered from older to younger chronostratigraphic occurrences, and comments on their updated status.

| Original reference | Identified taxa | Age | Horizon | Locality | Repository | Updated status |
|---|---|---|---|---|---|---|
| *Muñoz Ramírez et al., 2007* | *Squalus* sp. *Squatina* sp., *Cretorectolobus* sp., *Carcharias* sp., *Palaeohypotodus* sp., *Scapanorhynchus* sp., *Palaeogaleus* sp., *Galeorhinus* sp., *Paraorthacodus* sp., *Dasyatis* sp., *Rhinoptera* sp., ?Dasyatidae indet. | Paleocene | Cosmito Formation | Talcahuano, Región del Biobio | Museo Lajos Biró (Q.) | – |
| *Groz & Palma-Heldt, 2013* | *Squalus* sp., *Squatina* sp., *Carcharias* sp., *Palaeohypotodus* sp., *Scapanorhynchus* sp., *Paraorthacodus* sp., *Rhinoptera* sp., *Cretorectolobus* sp., *Palaeogaleus* sp., *Galeorhinus* sp., *Dasyatis* sp. | Paleocene | Cosmito Formation | Talcahuano, Región del Biobio | not indicated. Likely, Museo Lajos Biró (Q.) | review and stratigraphic assessment of samples described by *Muñoz Ramírez et al. (2007)* |
| *Rodriguez, Ward & Quezada, 2023* | *Paraorthacodus clarkii, Squalus minor, Squalus orpiensis, Centrophorus* sp., *Squatina prima, Anomotodon novus, Striatolamia striata, Carcharias* spp., *Sylvestrilamia teretidens, Odontaspis winkleri, Palaeohypotodus speyeri, Palaeohypotodus rutoti, Isurolamna inflata, Premontreia gilberti, Physogaleus secundus, Palaeogaleus vincenti, Hypolophodon sylvestris.* | late Paleocene | Pilpilco Formation | Puente Perales and Cerro San Martín, Región del Biobio | Museo Lajos Biró (Q.) | Review needed. This study suggest keeping under open nomenclature the following taxa due to dubious adscription based on a single sample: *Paraorthacodus* sp., *Squalus* sp., *Centrophorus* sp., *Squatina* sp., *Anomotodon* sp. *Carcharias* spp., *Sylvestrilamia teretidens, Odontaspis winkleri, Palaeohypotodus speyeri, Palaeohypotodus rutoti* and *Isurolamna inflata*. On the other hand, *Premontreia gilberti, Physogaleus secundus, Palaeogaleus vincenti* are easily distinguishable based on a single tooth; finally *Striatolamia striata* and *Hypolophodon sylvestris* are based in several samples. |
| *Fernández-Jiménez et al., 2016* | *Carcharias "hopei", Carcharias* sp., *Striatolamia macrota, Myliobatis* sp. | Eocene (undifferentiated) | Cosmito Formation | Coliumo, Región del Biobio | Museo Lajos Biró (Q.) | – |
| *Oliver-Schneider, 1936,* | *Odontaspis elegans* | Eocene (undifferentiated) | Lebu Group, without further resolution | Lota and Lebu river, Región del Biobio | unknown | unverifiable |
| *Oliver-Schneider, 1936* | *Odontaspis contortidens* | Eocene (undifferentiated) | Lebu Group, without further resolution | Lebu | unknown | unverifiable |
| *Suárez & Marquardt, 2003* | *Myliobatis* sp. | Eocene (undifferentiated) | Estratos de Algarrobo | Algarrobo, Región de Valparaíso | unknown | unknown repository; at least two genera of Myliobatoidea occur in Algarrobo *Otero (2024)* |
| *Suárez & Otero, 2010* | '*Isurolamna*' sp. | middle-upper Eocene | Millongue Formation | Loanco, Región del Maule | MNHN (SGO.PV.6635) | *Lethenia carranzaensis* sp. nov.; this study |
| *Otero, 2015* | '*Isurolamna*' sp., *Macrorhizodus praecursor* | middle-upper Eocene | Millongue Formation | Loanco, Región del Maule | not indicated then (MNHN; SGO.PV.6633) | not changed. Described by first time in this study |

*Paleobiographic relevance of the studied material*—The new material of *Palaeohypotodus* sp. adds to previous undifferentiated Paleocene (*Muñoz Ramírez et al., 2007*; *Groz & Palma-Heldt, 2013*) and late Paleocene records of the genus (*Rodriguez, Ward & Quezada, 2023*) in central Chile. Additional records of *Palaeohypotodus* in the WBP are known in lower Eocene units of the Austral Basin (*Otero & Soto-Acuña, 2015*) and Antarctica (*Long, 1992*; *Kriwet, 2005*; *Reguero et al., 2013*; *Kriwet et al., 2016*), although, the genus remains unreported in coeval units of New Zealand. Otherwise, in the northern hemisphere, *Palaeohypotodus* was restricted to the Danian-Thanetian of Europe, North America, central Asia and northern Africa (*Cappetta, 2012*). Thus, their current southeastern Pacific records (=central Chile) are coeval with those records from the northern hemisphere, but comparatively older than known high-latitude austral occurrences of *Palaeohypotodus*. These age differences likely represent an extension of the former Danian-Thanetian biochron of *Palaeohypotodus* (*Cappetta, 2012*) into the lower Eocene (in Antarctica and the Austral Basin), suggesting that the taxon was a relict in high latitudes of the southern hemisphere prior to its extinction.

Occurrences of the genus *Megasqualus* in the WBP have been reported in the Danian-Selandian of New Zealand (*Mannering & Hiller, 2008*), while northern records are known in the middle Paleocene-lower Eocene of Europe (*Cappetta, 2012* and references therein). The current material from central Chile represents the second known occurence of *Megasqualus* in the WBP.

On the other hand, the new occurrence of *Macrorhizodus praecursor* in middle-to-late Eocene levels of Loanco represents the northernmost occurrence of the taxon in the WBP. Their previous austral records were restricted to the Eocene of Antarctica (*Cione & Reguero, 1994*; *Reguero, Marenssi & Santillana, 2012*) and the Eocene of the Austral Basin in southern South America (*Otero et al., 2012*; *Otero et al., 2013*). A closely related lamnid taxon likely occurs in the Paleocene of New Zealand but the material was incomplete for an accurate taxonomical referral (*Mannering & Hiller, 2008*: fig 15). Otherwise, the genus *Macrorhizodus* had a widespread distribution along Europe, Atlantic Africa and Asia between the lower Eocene - lower Oligocene (*Cappetta, 2012* and references therein).

In addition, the description of *Lethenia carranzaensis* sp. nov. represents the first occurrence of the genus in the southern hemisphere. Previous reports are known in the lower Oligocene of Belgium (*Baut & Génault, 1999*), France and Kazakhstan (*Cappetta, 2012*), and the upper Oligocene of Europe (*Cappetta, 2012*).

Finally, the eventual presence of and indeterminate hexanchid may represent its current oldest record along the southeastern Pacific. Austral record of hexanchids can be tracked back to the Late Jurassic of New Zealand (*Cappetta & Grant-Mackie, 2018*). There is a major gap during the austral Cretaceous. The group was frequently reported in the James Ross Basin since the Late Cretaceous until the Eocene (*Grande & Chatterjee, 1987*; *Richter & Ward, 1990*; *Long, 1992*; *Cione & Reguero, 1994*; *Kriwet et al., 2006*; *Reguero, Marenssi & Santillana, 2012*; *Engelbrecht et al., 2017*). Hexanchids are also known in the Paleocene of New Zealand (*Mannering & Hiller, 2008*). In South America, the oldest hexanchid records in the Austral (=Magallanes) Basin are known in the Maastrichtian of Calafate Lake (*Bogan, Agnolin & Novas, 2016*). Younger records are relatively frequent and broadly distributed

along the entire basin during the Eocene (*Otero et al., 2012*; *Otero et al., 2013*). On the contrary, records of hexanchids along the southeastern Pacific are known since the Miocene and onwards (*Long, 1993*; *Carrillo-Briceño et al., 2013*; *Suárez, 2015*; *Chávez-Hoffmeister & Villafaña, 2023*), but its local presence before the Miocene remained unreported until now.

*Palaeogeography*—The Paleocene chondrichthyan diversity previously described from central Chile (*Muñoz Ramírez et al., 2007*; *Groz & Palma-Heldt, 2013*; *Rodriguez, Ward & Quezada, 2023*) includes several genera with widespread representation along the WBP or even being cosmopolitan (*i.e.,* genera *Squatina*, *Carcharias*, *Palaeohypotodus*, *Scapanorhynchus*, *Galeorhinus*, *Paraorthacodus*, *Dasyatis*, *Rhinoptera*, *Centrophorus*, *Anomotodon*, *Striatolamia*, *Odontaspis* and *Hypolophodon*). Exceptions are the single record of a worn tooth of *Cretorectolobus* sp. (*Muñoz Ramírez et al., 2007*) that might belong to reworked material from older units, considering its known bichron restricted to the Hauterivian-Maastrichtian (*Cappetta, 2012*).

The new Paleocene-lower Eocene records of *Palaeohypotodus* sp., *Megasqualus* sp. and Hexanchidae indet. extend their known distribution in the WBP. The same applies for the new record of *Macrorhizodus praecursor*, also present in austral Eocene WBP localities. On the contrary, the reported presence of *Palaeogaleus*, *Physogaleus*, and *Premontreia* in the upper Paleocene of central Chile (*Rodriguez, Ward & Quezada, 2023*) supports an early oceanic connection with the North Atlantic. While these taxa remain unreported in high latitudes of the WBP, the three genera are known in Europe, North America, north and west Africa, and the Near East (*Cappetta, 2012*). In addition, the new record of the genus *Lethenia* (represented by *Lethenia carranzaensis* sp. nov.), complements a similar scenario during the late Eocene-early Oligocene. Previous to this research, the genus *Lethenia* was exclusively reported in the Oligocene of Europe and central Asia (*Baut & Génault, 1999*; *Cappetta, 2012*), remaining unreported in any coeval locality of the southern hemisphere.

This pattern of an Atlantic influence into the WBP, likely beginning in the late Paleocene and extending into the early Oligocene, is reinforced by the similar distribution of the billfish clade Blochiidae, a group firstly described in the Eocene of Italy (*Fierstine & Monsch, 2002*), but later shown to be present in the middle-to-late Eocene of central Chile (particularly in Loanco, the locality studied herein), represented by an endemic genus and species (*Otero, 2019*). The occurrence of these taxa suggest an emerging picture for a temporal and latitudinal declination of the WBP. While endemic taxa indeed occur (*i.e., Lethenia carranzaensis*, *Loancorhynchus catrillancai*: *Otero, 2019*; this study), their closest relatives are known in the North Atlantic.

This faunal interchange between the southeastern Pacific and the north Atlantic plus its chronostratigraphic occurrences, were expectable, considering the composition of the Neogene chondrichthyan fauna known in central and northern Chile (*Carrillo-Briceño et al., 2013*; *Suárez, 2015*; *Chávez-Hoffmeister & Villafaña, 2023*), which contrasts in diversity to the Eocene assemblages of southernmost Chile and Antarctica (*Long, 1992*; *Kriwet, 2005*; *Kriwet et al., 2016*; *Reguero, Marenssi & Santillana, 2012*; *Reguero et al., 2022*; *Otero et al., 2012*; *Otero et al., 2013*). Previous to this research, there was no evidence of a latitudinal segment in the southeastern Pacific, neither a time span for the beginning of such faunal turnover.

The faunal interchanges above commented, occurred in a moment of major global geographic and climatic dynamics which especially affected the southern hemisphere by the opening of the Tasmanian Gateway, the deepening of the Drake Passage (*Lawver & Gahagan, 2003*; *Case, 2007*), the extension of the Antarctic ice sheet, plus variations in the Antarctic ocean circulation patterns (*Goldner, Herold & Huber, 2014*) and sea temperature (*Kennedy et al., 2015*). Although sparse, the new material described in this contribution sheds the first insights on the marine vertebrate faunal changes coupled to these global dynamics along the southern Pacific coast of South America.

## CONCLUSIONS

New Paleogene chondrichthyan remains from central Chile are here studied. The oldest studied sections correlates with the Paleocene-lower Eocene Pilpilco Formation, which includes the northernmost occurrences along the Pacific of *Palaeohypotodus* sp., *Megasqualus* sp. and Hexanchidae indet. The new record of *Megasqualus* is the second known in the WBP, while the material referred to Hexanchidae indet. represents the earliest known evidence of the group in the southeastern Pacific.

Two additional sections of a second younger unit (Millongue Formation) have yielded material referable to lamnid sharks, represented by *Macrorhizodus praecursor* (Leriche), and by *Lethenia carranzaensis* sp. nov., described herein. The new record of *Macrorhizodus* represents its northernmost occurrence in the WBP, although this was a cosmopolitan taxon during the Eocene. On the contrary, the new record of *Lethenia* represents its first occurrence in the southern hemisphere. Its distribution has a similar pattern previously recognized in coeval billfishes of the clade Blochiidae. Moreover, both taxa share a similar distribution with the chondrichthyan genera *Palaeogaleus*, *Physogaleus*, and *Premontreia*, previously reported in the upper Paleocene of central Chile. All these taxa remain unreported in higher latitudes of the WBP but they are known in the Atlantic realm during the same time span, suggesting that a southeastern Pacific-north Atlantic faunal interchange occurred at least since the upper Paleocene until the lower Oligocene. Such interchange could represent the starting point of the WBP declination as a recognizable bioprovince, prior to the establishment of typical Neogene marine vertebrate faunas along the southeastern margin of the Pacific Ocean.

## ACKNOWLEDGEMENTS

Thanks to S Soto (Universidad de Chile), and C Salazar (U. del Desarrollo, Chile) A Rubilar (Sernageomin) and FA Mourgues (Terra Ignota), for their participation and support in different field work between 2008 and 2015, which allowed the recovery of part of the material presented here. The author thanks M Reguero (Universidad de La Plata), EV Popov (Saratov State University, Russia) O Kovalchuk (Academy of Sciences, Ukraine), and one anonymous reviewer, for the valuable comments that improved the original draft of this work.

### Funding

This research and its final stage was supported by the Nucleo Milenio EVOTEM (ANID-MILENIO-NCN2023-025). Early parts of this research were supported by the projects ARTG-04 Anillo Antártico ANID-PBCT (2007-2010), ACT-105 Anillo Antártico, ANID-Chile (2010-2012), and Anillo ACT-172099, ANID-Chile (2018-2020). The funders had no role in study design, data collection and analysis, decision to publish, or preparation of the manuscript.

### Grant Disclosures

The following grant information was disclosed by the author:
Nucleo Milenio EVOTEM: ANID-MILENIO-NCN2023-025.
ARTG-04 Anillo Antártico ANID-PBCT (2007-2010).
ACT-105 Anillo Antártico, ANID-Chile (2010-2012).
Anillo ACT-172099, ANID-Chile (2018-2020).

### Competing Interests

The authors declare there are no competing interests.

### Author Contributions

- Rodrigo A. Otero conceived and designed the experiments, performed the experiments, analyzed the data, prepared figures and/or tables, authored or reviewed drafts of the article, and approved the final draft.

### Data Availability

The specimen is housed at Museo Nacional de Historia Natural, Santiago, Chile (SGO.PV.6635).

### New Species Registration

The following information was supplied regarding the registration of a newly described species:

Publication LSID: urn:lsid:zoobank.org:pub:FF2D99EE-7705-4F96-A7F4-53D18A520EF9
Species name: urn:lsid:zoobank.org:act:012B0463-22D3-4A40-916A-F005532BD09B.

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
