# Peer review of "New Paleogene records of cartilaginous fishes (Chondrichthyes) from central Chile, including the oldest lamnid diversity from the southeastern Pacific"

_PeerJ, doi:10.7717/peerj.19996_

## Round 0.1 · original submission · Major Revisions

I agree with the reviewers in considering your work to be of scientific interest and potentially suitable for publication in PeerJ. However, I think that the manuscript requires major revisions in order to meet the journal's standards.

I kindly ask that you carefully address all of the reviewers’ comments and suggestions in your revised manuscript, paying particular attention to the detailed feedback provided by Reviewer 2, whose concerns are especially important for the improvement of your work. Please include a point-by-point response to all reviewers' comments with your resubmission, clearly outlining how each issue has been addressed.

We look forward to receiving your revised manuscript.
Best regards,
Paula Bona
Academic Editor
PeerJ

·

Basic reporting

It is a very interesting and well-written paper. It provides useful information about the Paleogene chondrichthyan diversity from central Chile.

148-153 Some comment about thAge of the Outcrops Studied Here, stratigraphic occurrence of Palaeohypotodus.
Long (1992) reported Palaeohypotodus rutoti from Cucullaea I Allomember (TELM 5) of La Meseta Formation, Seymour (Marambio) Island. The weddellian occurrence of Palaeohypotodus stratigraphically is restricted to the middle Eocene of La Meseta Formation in Seymour (Marambio) Island (Charnelli et al., 2024).
I suggest to add Charnelli et al., 2024 to the references of this paper.

Charnelli M, Gouiric-Cavalli S, Reguero M A, et al. Middle Eocene chondrichthyan fauna frohe referencesm Antarctic Peninsula housed in the Museo de La Plata, Argentina. Adv Polar Sci, 2024, 35(1): 14-47, doi: 10.12429/j.advps.2023.0035

Experimental design

No comments

Validity of the findings

No comment

Reviewer 2 ·

Basic reporting

The manuscript is generally well-written in professional English. However, I read both American and British English words, so it's important to choose one and stay consistent throughout the manuscript.

In my opinion, the introduction is too short and too concise to fully understand the scientific context of this study. I would appreciate more information on previous analyses of the vertebrate faunas of the studied region. Furthermore, Table 1, summarising the previously identified shark species, should be included in the introduction. A major structural issue is in the last paragraph, in which the results/conclusions are announced before any further analysis/discussion. This last paragraph should explain the scientific question(s) and hypothesis(es) this article aims to test. A more thorough explanation of the scientific context would help the author formulate it.

The overall structure of the article is consistent with a systematic study, but I have highlighted several issues in the systematic palaeontology section. Since most taxa are identified at the generic level, I suggest adding additional sections (type locality, included species, diagnosis, remarks if necessary) for the identified genera.
Another issue concerns the use of a "Remarks" section after the description of the material. I believe this term is inappropriate, as in this section the author analysed and compared the morphology of the taxa with comparative material. This section should therefore be identified as a comparison section.

The figures are good and accurate, and I propose only a few minor changes in Figure 1. In addition, it might be useful to add a fourth figure to illustrate on a palaeo-map the distribution of the taxa discussed.

Experimental design

This work presents the description of a new fossil shark fauna, as well as discussions on the palaeobiogeography of these sharks in the Southern Hemisphere. It is therefore in accordance with the journal's aim and scope.

The introduction is too short, making it difficult to assess the scientific question the author wants to answer/test. Furthermore, while I understand the novelties brought by this study (extensive geographical and temporal distribution of some Lamnids in the Paleocene-Eocene), I do not understand the lack of knowledge that "justifies" this study. I recommend that the author provide a more complete introduction to better define the scientific context of this study.

I understand that the descriptions provided are brief due to the scarcity of the material studied. However, I believe that the morphological comparisons are insufficient, both in quality and quantity, to support the taxonomic assignment. I would appreciate a more coherent structure, starting with the justification for the assignment to family and ending with the more precise taxonomic level (species or genus). Here, the author limits himself to a brief comparison with the previous taxonomic assignment to justify the systematic changes.
Furthermore, as the author points out in the "Discussion" section, I do not believe that one or two teeth are sufficient to identify a specific species or genus, especially given the potentially high complexity of shark heterodonty. I recommend that the author reevaluate the taxonomic classification by providing a more comprehensive comparative section.
For newly established species, the author should include a full description of all material, even that which is not diagnostic for lamnid sharks at the species level.

No new methods are described here, and this work corresponds to the standard protocol of palaeontological studies, even considering some lacks in the systematic palaeontology section.

Validity of the findings

This work illustrates the new material well; replication of the taxonomic identification could therefore be easily performed by any other researcher. The results are therefore novel and well established in the text. I again recommend the addition of a fourth figure to illustrate the updated palaeobiogeography of the taxa studied.

My understanding is that the material is housed in a public institution, which allows for long-term preservation of the material and ensures its availability for any future revision work.

Additional comments

The author describes a new fauna of lamnid sharks from several localities in central Chile, spanning the period from the Early Paleocene - Early Eocene and from the Late Eocene - Early Oligocene. The identification of five taxa, including one new species, allows to discuss the palaeobiogeography of lamnid sharks in the Southern Hemisphere, and more specifically in the Weddellian Biogeographic Province. Thus, faunal connections with the North Atlantic are confirmed during the Late Eocene - Early Oligocene period, and stratigraphic occurrences of lamnid sharks are geographically extended in the southwest Pacific during the Late Paleocene period.

The author certainly describes a new fauna and offers logical palaeobiogeographic interpretations. However, the description and morphological comparisons provided in the manuscript are too concise to truly assess the accuracy of the taxonomic identification. I strongly recommend that the author conduct a more in-depth morphological analysis to support the proposed systematics.
Another essential point concerns the structure of the manuscript, especially the introduction. The author should provide a more detailed description of the scientific context of the study so that the scientific question and the hypotheses tested in this study are more easily understood by readers.

Additionally, I have added several comments directly in the text (annotated PDF document).

I therefore consider this work to be of scientific interest for publication in PeerJ, but it requires several major revisions to meet the journal's standards.

Annotated reviews are not available for download in order to protect the identity of reviewers who chose to remain anonymous.

·

Basic reporting

The manuscript subjected for review presents the results of the study of a new Paleogene chondrichthyan assemblage from central Chile, including the description of a new species - Lethenia carranzaensis - and other oldest lamnid records from the southeastern Pacific. The text is generally well written although the meaning of some sentences is unclear (see below). I've made a number of corrections directly in the PDF file but the text should be proofreaded by a native speaker for clarity. Literature references are adequate, and all of the are needed. The structure of the article is fine being similar to other systematic papers previously published in the journal. The figures are of superb quality, they successfully complement the content of the article.

Experimental design

The research presented in the text fits well within the Aims and Scope of PeerJ. Research questions raised by the author are well defined and meaningful. This study fills an important gap in the knowledge about early Paleogene chondrichthyan assemblages of the southeastern Pacific. The investigation has been conducted rigorously and the methods used are desribed in detail.

Validity of the findings

This submission represents a new important contribution to the field, and when published it will be useful for a vertebrate paleontological community. Conclusions are in general well stated and supported by a massive of data prepsented in the text.

Additional comments

There are a few additional suggestions which I would encourage the authors to address when preparing the revised version of the manuscript.

1. Please avoid mixing chronostratigraphic (e.g., lower, upper) and geochronological terms (e.g., early, late) in the text: the former can be used in relation to deposits while the age of fish fossils or paleoassemblages should be accompanied exclusively with the latter.
2. It is unclear what does "dental pieces" mean, it should be clarified elsewhere.
3. The meaning of some sentences is not clear, they should be rewritten.
4. The diagnosis of a new species Lethenia carranzaensis is very short and should be supplemented with more details.

·

Basic reporting

Clear and unambiguous, professional English used throughout.
There are some minor comments on the text:
Title: New Paleogene records of chondrichthyan fishes (Elasmobranchii) from central Chile...
1. p.1, line 29: ‘dental pieces of chondrichthyans’ – the term ‘dental pieces’ is not frequently used; more useful and common terms are ‘teeth’ or ‘teeth fragments’;
2. p. 11, line 351, 357: ‘two blunt triangular lateral cusplets, the larger cusplet is blunt’ – The cusplets of Lethenia carranzaensis teeth can be described as triangular, relatively low and broad, but have a well-formed apex, therefore the term ‘blunt’ is not fit their morphology very well. For example, typical blunt lateral cusplets with a rounded outline (lacking a sharp apex) are characteristic for many lateral teeth of Striatolamia, and some teeth of Sylvestrilamia;
3. p. 11, line 355: ‘anterior teeth with straight cusp in lateral and profile views’ – The difference between lateral and profile views is not clearly defined in the text. Maybe labial/lingual and profile views were meant?
Literature references, sufficient field background/context provided.
The paper provides needed field background, geological information, the context of previous records of Paleogene chondrichthyans in the region. The literature reference list includes mostly relevant papers, but some corrections should be done:
1. p.5, line 165: ‘Muñoz-Cristi, 1958’ – there is only reference for Muñoz-Cristi, 1968 in the reference list;
2. p. 17, line 552: ‘Case, 2007’ – this paper absent in the reference list;
3. p. 21, line 683: ‘Herman J. 1982. Die Selachier…’ – The genus Megasqualus was described in another article, the correct reference is Herman, J. 1982. Additions to the fauna of Belgium. 6. The Belgian Eocene Squalidae. Tertiary Research, 4(1), 1–6;
4. p. 25, line 805: ‘Winkler TC. 1874. Mémoire sur des dents de poissons du terrain bruxellien…’ –Megasqualus orpiensis and Palaeohypotodus rutoti were originally described in Winkler, T.C. 1874. Mémoire sur quelques restes de poissons du système heersien. Archives du Musée Teyler, 4(1), 1–15.

Professional article structure, figures, tables. Raw data shared.
The comments on the figures are following:
1. in the fig.1 the stratigraphic columns for points C and D are swapped places: as stated in the text, Macrorhizodus teeth were found in the point C and Lethenia - in the point D (the columns show an opposite situation). Also, in the stratigraphic column for point C the Pilpilco Formation is shown, while in the text these deposits referred to the Millongue Formation. Noted contradictions between the figure and the text should be corrected;
2. in the fig. 2 the picture of Hexanchiform tooth is not very good quality and described ‘longitudinal striations over the lingual face of the cusp and the radial striations associated to the cutting edges’ are not visible. The better photograph of this tooth is recommended. In the figure explanation for Macrorhizodus (Fog. 2F-G) there is a misprint (‘upper inferior tooth’ instead anterior tooth), and the explanation for fig.2E is missing.
3. Fig. 3 displays the most unique specimen described in the paper – the teeth set, cartilages and vertebras of one individual of Lethenia. The additional graphics of this rare specimen will be useful. The figure 3 lacks the picture of this specimen before the preparation or the scheme (drawing) showing the position of teeth and cartilages in the matrix. Now the placement of these elements before their extraction from the deposits is unclear. Moreover, it would better mark different elements (palatoquadrates, Meckel's cartilages) in the figure (current figure explanation as ‘cartilaginous fragments of the jaw’ is too laconic).
Some improvements are recommended for the structure of ‘Systematic’ section. For Megasqualus (p.6, line 199) and Macrorhizodus (p. 10, line 310) the information about type species is placed after the synonymy section, but probably the better (and more classical) way to put it just below the line with ‘genus name’ (as it is done for Palaeohypotodus and Lethenia). Also, the paragraph with ‘Etymology’ for Lethenia carranzaensis (p. 11, line 353) would better placed before the ‘diagnosis’; thereby the ‘diagnosis’ will be followed by the ‘description’.

Self-contained with relevant results to hypotheses.
‘no comment’

Experimental design

Original primary research within Aims and Scope of the journal.
‘no comment’

Rigorous investigation performed to a high technical & ethical standard.
‘no comment’

Methods described with sufficient detail & information to replicate.
‘no comment’

Validity of the findings

Impact and novelty not assessed. Meaningful replication encouraged where rationale & benefit to literature is clearly stated.
‘no comment’

All underlying data have been provided; they are robust, statistically sound & controlled
‘no comment’

Conclusions are well stated, linked to original research question & limited to supporting results.
The teeth described as Hexanchidae indet. and Palaeohypotodus sp. are poorly preserved, and their identification should be regarded tentative. Therefore, the parts of discussion concerning these teeth (in ‘Paleobiographic relevance of the studied material’ and ‘Palaeogeography’) should be presented in a more hypothetical way (e.g. ‘(?)Palaeohypotodus sp. probably adds to previous undifferentiated Paleocene and late Paleocene records of the genus…’, ‘the possible presence of and indeterminate hexanchid may represent its current oldest record along the southeastern Pacific…’).
The information about oceanic and climatic events, which noted in the last paragraph of ‘Discussion’, is not necessary, since it cannot be convincingly linked with rare chondrichthyans finds.
Except for the comments noted above, the conclusions are supported by better preserved and identifiable material.

Additional comments

The new species Lethenia carranzaensis is based on complete material including teeth of different positions, cartilages of mandibular arch and vertebras. However, the description deals only with teeth, while cartilages and vertebras are not described. These skeletal elements of the holotype should be described as well as the teeth. The diagnosis of the new species could be improved by adding more features: e.g. L. carranzaensis has complete cutting edges reaching the base of the crown, more subrectangular root lobes (especially in anterior and anterolateral teeth), and lateral cusplets, which is not curved lingually, in contrast with incomplete cutting edges, rounded root lobes and curved lateral cusplets in L. vanderbroecki.
Other additional comments on the text are attached in PDF.

---

## Round 0.2 · Minor Revisions

Your work is nearly ready; however, there are still a few issues that need to be addressed before it can move forward.

In particular, I recommend that you carefully consider the suggestions made by Reviewer 1, especially the request to base taxonomic identifications strictly on morphological comparisons, providing a more in-depth comparison to justify the taxonomic attributions (even at the generic level).

If you decide not to implement any of these suggestions, you must provide a clear and well-founded justification, which must be explained directly in the manuscript and in the rebuttal letter.

Reviewer 2 ·

Basic reporting

I appreciated the changes made by the authors according to my previous comments and suggestions. However, I have still concern about some of the taxonomic attributions et the fact no change was provided to asnwer to my previous comments.

The author identifies specimens of Squalidae and Odontodaspidae at the generic level based on one or two teeth, even though he claimed the opposite, including what he wrote in the first version of his manuscript and in his reply. Neither in his reply nor in the R1 version did I find evidence that this precaution is unnecessary for Megasqualus and Palaeohypotodus. I understand it better for Macrorhizodus, but the justification remains largely superfluous in the text. Consequently, my previous comment still stands. Once again, the comparisons are too confusing and require sentences clearly explaining why a given specimen is assigned first to a given family, then to a given genus, on the basis of morphological characteristics.

If the goal is to reevaluate the systematic assignments of certain specimens, any mention of a previous assignment or the local paleontological context in the comparison should be removed in order to propose an identification based solely on morphology. References to previous work and paleontological context should be reserved for the first part of the discussion. Furthermore, the addition of diagnoses for generic-level identifications would be very helpful in enabling readers to evaluate the generic attribution proposed here.

To resume, I highly recommend providing a more in-depth comparison to justify the taxonomic attribution, alongside the diagnoses of the different discussed genera. I also provided some other comments directly in the pdf of the revised version.

Experimental design

Nothing to add here.

Validity of the findings

Nothing to add here.

Additional comments

Nothing to add here.

Annotated reviews are not available for download in order to protect the identity of reviewers who chose to remain anonymous.

·

Basic reporting

Clear and unambiguous, professional English used throughout.

Experimental design

Original primary research within Aims and Scope of the journal.

Validity of the findings

Conclusions are well stated, linked to original research question & limited to supporting results.

Additional comments

The authors considered my suggestions and addressed them accordingly in the text. I have no additional comments and recommend accepting the revised submission as is.

---

## Round 0.3 · accepted · Accept

I confirm that the authors have thoroughly addressed all of the reviewers' comments and revised the manuscript accordingly. I concur with the recommendation for acceptance provided by one of the reviewers. Although the second reviewer declined to assess the revised manuscript, I am satisfied that their previous comments and observations have been fully addressed and appropriately answered by the authors in the final version. In my opinion, the manuscript meets the necessary scientific and editorial standards and is ready for publication.